# UNIMC: Taming Diffusion Transformer for Unified Keypoint-Guided Multi-Class Image Generation

**Qin Guo** [1]  **Ailing Zeng** [2]  **Dongxu Yue** [3]  **Ceyuan Yang** [4]  **Yang Cao** [1]  **Hanzhong Guo** [5]  **Fei Shen** [6]  **Wei Liu** [2]  **Xihui Liu** [5]  **Dan Xu** [1]

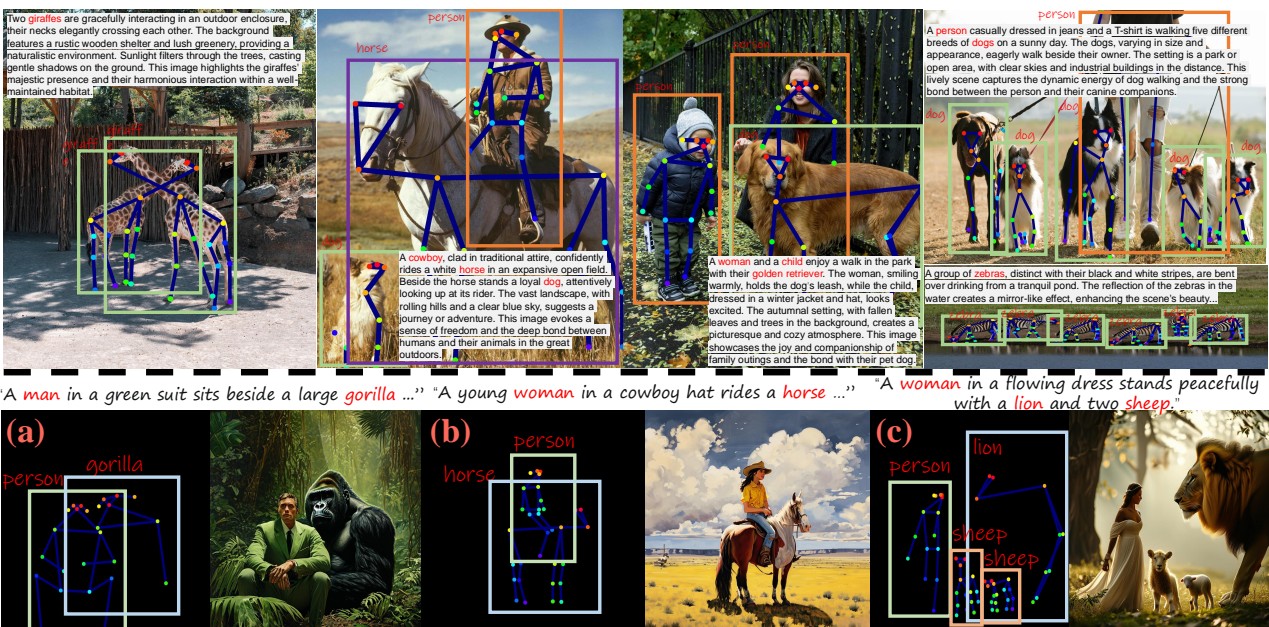

Figure 1. **Top:** We establish `HAIG-2.9M`, a large-scale, high-quality, and highly diverse dataset with joint keypoint-level, instance-level, and densely semantic annotations for both humans and animals. **Bottom:** Based on `HAIG-2.9M`, we design UNIMC, a controllable DiT-based framework for keypoint-guided image generation, especially for multi-class (e.g., (a), (b), (c)) and heavy occlusion scenarios (e.g., (a), (b)). The bottom part of the figure showcases samples generated by UNIMC.

## Abstract

Although significant advancements have been achieved in the progress of keypoint-guided Text-to-Image diffusion models, existing mainstream keypoint-guided models encounter challenges in controlling the generation of more general non-rigid objects beyond humans (*e.g.*, animals). Moreover, it is difficult to generate multiple overlapping humans and animals based on keypoint controls solely. These challenges arise from two main aspects: the inherent limitations of existing controllable methods and the lack of suitable datasets. First, we design a DiT-based framework, named UNIMC, to explore unifying controllable multi-class image generation. UNIMC integrates instance- and keypoint-level conditions into compact tokens, incorporating attributes such as class, bounding box, and keypoint coordinates. This approach overcomes the limitations of previous methods that struggled to distinguish instances and classes due to their reliance on skeleton images as conditions. Second, we propose `HAIG-2.9M`, a large-scale, high-quality, and diverse dataset designed for keypoint-guided human and animal image generation. `HAIG-2.9M` includes 786K images with 2.9M instances . This dataset features extensive annotations such as keypoints, bounding boxes, and fine-grained captions for both humans and animals, along with rigorous manual inspection to ensure annotation accuracy. Extensive experiments demonstrate the high quality of `HAIG-2.9M` and the effective-

[1]The Hong Kong University of Science and Technology [2]Tencent [3]Peking University [4]The Chinese University of Hong Kong [5]The University of Hong Kong [6]National University of Singapore. Correspondence to: Dan Xu <danxu@cse.ust.hk>.

*Proceedings of the 42nd International Conference on Machine Learning*, Vancouver, Canada. PMLR 267, 2025. Copyright 2025 by the author(s).

ness of UNIMC, particularly in heavy occlusions and multi-class scenarios.

## 1. Introduction

In recent years, diffusion models (Sohl-Dickstein et al., 2015; Ho et al., 2020) have gained significant attention due to their remarkable performance in image generation. The generation of non-rigid objects, such as humans and animals, is vital across various domains, including animation (Zhang et al., 2024; Hu et al., 2023), artistic creation (Midjourney, 2024), and animal-related perception tasks (Yu et al., 2021; Yang et al., 2022; Shooter et al., 2021). While class-conditioned (Dhariwal & Nichol, 2021) and text-conditioned (Rombach et al., 2022; Ramesh et al., 2022) image generation has achieved great success, more controllable generation is desirable, especially controlling the structure of non-rigid objects like humans and animals. Previous works on keypoint-guided human image generation (Zhang et al., 2023; Ju et al., 2023b; Liu et al., 2024b) have utilized skeleton images as structural conditions. However, it is difficult for those methods to generate multi-class, multi-instance images of humans and/or animals, especially in scenarios with overlapping instances.

We identify two primary reasons for this challenge: **1) Conditional formulation limitations**: Most methods plot keypoints on images as control signal (Zhang et al., 2023; Ju et al., 2023b; Liu et al., 2024b), as shown in Fig. 2. **2) Dataset limitations**: Most existing keypoint-annotated datasets focus on humans (Lin et al., 2014; Ju et al., 2023a), with only a few annotating animals (Yu et al., 2021; Yang et al., 2022). None provide keypoint annotations for images featuring both humans and animals. These datasets are designed for perception tasks, lacking the quantity and quality needed for current generative models.

Such skeleton image conditions face two main issues: **1) Class binding confusion**. The keypoints plotted on the image lack category information. For example, we cannot distinguish if the keypoints should depict a cat or a dog. **2) Instance binding confusion.** For images with overlapping instances, it is challenging to distinguish which instance a keypoint belongs to in the overlapping regions. Moreover, variations in skeleton rendering (*e.g.*, color, line width) further hinder the generation quality. Consequently, using skeleton images as conditions leads to suboptimal performance for multi-class, multi-instance generation.

To overcome the condition formulation limitations, we propose UNIMC, a unified Diffusion Transformer (DiT)-based (Peebles & Xie, 2023) framework designed for keypoint-guided generation of humans and animals. Specifically, we use each instance's class name, bounding box and keypoint coordinates instead of skeleton images as the condition signal, thus addressing class binding confusion and instance binding confusion. We then propose *unified keypoint encoder*, a lightweight keypoint encoder that encodes keypoints of different species into a universal token representation space, avoiding the need for training separate encoders for each species and enhancing both training and inference efficiency. Next, we introduce *timestep-aware keypoint modulator*, where keypoint tokens are injected into the DiT-based generator to achieve keypoint-level control for multi-class, multi-instance image generation.

To overcome the limitations of the dataset, we propose `HAIG-2.9M`, which includes 786K images with an average aesthetic score (christophschuhmann, 2023) of 5.91, covering 31 species classes, and annotated with 2.9M instance-level bounding boxes, keypoints, and captions, averaging 3.66 instances per image. Following (Yang et al., 2022), these 31 classes can be further classified into 15 animal families following taxonomic rank to facilitate the evaluation of inter-species and inter-family generalization ability of generative models. Among these, nearly half of the images contain animals, and one-quarter of the images feature both humans and animals. Specifically, to ensure data quality and diversity, we crawl 460K images from four high-quality **non-commercial** data websites (Pexels, 2024; Pixabay, 2024; stocksnap, 2024; unsplash, 2024) and filter 2.07M images from four high-quality datasets (Schuhmann et al., 2024b;a; Sun et al., 2024; Kirillov et al., 2023). We annotate bounding boxes, human and animal keypoints, and captions using a selection of state-of-the-art (SOTA) models for each annotation type. First, we label 5% of randomly sampled data, then manually evaluate the annotations, selecting the top-performing model for each type.

To summarize, our main contributions are three-fold:

- We introduce a unified DiT-based framework, UNIMC, designed for multi-class, multi-object image generation using keypoint conditions. This framework includes the *unified keypoint encoder* that encodes keypoints of different species into the shared representation space and the *timestep-aware keypoint modulator* for effective keypoint-level control.

- We propose a unified human and animal image dataset, `HAIG-2.9M`, for keypoint-guided image generation. This large-scale and diverse dataset includes 786K images and 2.9M instances with comprehensive annotations such as keypoints, bounding boxes, and captions, covering 31 species classes and 15 animal families, facilitating multi-class, multi-object image generation.

- Extensive experiments demonstrate the high quality of `HAIG-2.9M` and the effectiveness and efficiency of UNIMC.

*Table 1.* **Comparison of Mainstream Human and Animal Keypoint-Image Paired Datasets** (Yang et al., 2022; Lin et al., 2014; Ju et al., 2023a) and **General T2I Datasets** (Schuhmann et al., 2024b;a; Sun et al., 2024; Kirillov et al., 2023) **with HAIG-2.9M.** For COCO, we only calculate statistics of images that contain human keypoint annotations. For general T2I datasets, we exclusively report the scene types and the number of images.

| Dataset | Scene Type | Image | Instance | Keypoints Human | Keypoints Animal | Average Aesthetic Score | Average Caption Length | Average Resolution |
|---|---|---|---|---|---|---|---|---|
| JourneyDB (Sun et al., 2024) | Artistic | 4,738,554 | \ | × | × | \ | \ | \ |
| SA1B (Kirillov et al., 2023) | Real World | 11,000,000 | \ | × | × | \ | \ | \ |
| LAION-PoP (Schuhmann et al., 2024b) | Diverse | 600,000 | \ | × | × | \ | \ | \ |
| LAION-Aes-V2 (Schuhmann et al., 2024a) | Diverse | 625,000 | \ | × | × | \ | \ | \ |
| COCO (Lin et al., 2014) | Real World | 58,945 | 156,165 | ✓ | × | 4.981 | 10.5 | 578 × 484 |
| Human-Art (Ju et al., 2023a) | Artistic | 50,000 | 123,131 | ✓ | × | 5.409 | 10.5 | 1297 × 1298 |
| APT36K (Yang et al., 2022) | Real World | 36,000 | 53,006 | × | ✓ | 4.516 | \ | 1920 × 1080 |
| **HAIG-2.9M (ours)** | Diverse | **786,394** | **2,874,773** | ✓ | ✓ | **5.914** | **77.48** | **1775 × 1602** |

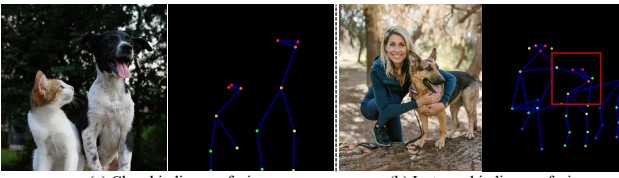

(a) Class binding confusion      (b) Instance binding confusion

*Figure 2.* Skeleton image conditions face two main issues: **(a) Class binding confusion**: difficult to distinguish classes from skeleton images alone; **(b) Instance binding confusion**: challenging to distinguish keypoints of overlapping instances under occlusions (*e.g.*, parts in red box).

## 2. Related Work

**Keypoint-Guided Image Generation.** Controllable image generation is a critical research direction, evolving from early Generative Adversarial Networks (Creswell et al., 2018; Ma et al., 2017) and Auto-Encoders (Ren et al., 2020; Esser et al., 2021) to the recent diffusion models (Zhang et al., 2023; Mou et al., 2024; Qin et al., 2024; Zhao et al., 2024; Li et al., 2023). Keypoint-Guided image generation is a vital branch of controllable generation, with recent diffusion model-based works like ControlNet (Zhang et al., 2023), T2I-Adapter (Mou et al., 2024), HumanSD (Ju et al., 2023b), and HyperHuman (Liu et al., 2024b) enabling high-quality controllable human synthesis. These methods typically input skeleton images as control signal, but this form faces **class binding confusion** and **instance binding confusion** as discussed in Sec. 1 and shown in Fig. 2, leading to significant information loss. GLIGEN (Li et al., 2023) initially explored using keypoint positions for human image generation, and subsequent works (Wang et al., 2024; Zhou et al., 2024) adopted similar architectures for instance-level control. However, previous methods focused exclusively on keypoint-guided human generation, overlooking more general applications such as keypoint-guided animal generation. As demand for flexible control in image generation grows, especially given the importance of animals, precise keypoint-based control over both human and animal generation becomes crucial. Additionally, varied keypoint definitions across classes introduce semantic and structural gaps, complicating unified generation. In response, we pro-

pose UNiMC, which encodes keypoints of different species into a unified representation, enabling keypoint-level control within a single framework.

**Diffusion Transformer.** In this work, we use DiT as our backbone. For details on DiT, see Apx. B.

**Datasets for Human and Animal Image Generation.** Large-scale and high-quality datasets are crucial for image generation (Chen et al., 2024c). However, existing human-centric datasets often encounter issues such as low quality (Lin et al., 2014; Zheng et al., 2015; Han et al., 2018), limited diversity (Fu et al., 2022; Liu et al., 2016), limited size (Ju et al., 2023a; Gong et al., 2017), lack of open access (Liu et al., 2024b), a focus on few person scenes (Qin et al., 2024; Li et al., 2024a), and only skeleton image annotations (Qin et al., 2024). Similarly, animal-centric datasets (Yu et al., 2021; Yang et al., 2022; Ng et al., 2022), most designed for perception tasks, also suffer from very poor quality, small size, and low diversity. Furthermore, none of these datasets offer keypoint annotations for both humans and animals co-existing, hindering the development of keypoint-guided human and animal image generation. In response, we propose HAIG-2.9M, as shown in Tab. 1, which contains more than ten times the number of images and instances compared to commonly used datasets (Yang et al., 2022; Lin et al., 2014), with significantly improved aesthetic scores. It includes comprehensive annotations for both humans and animals, along with high-quality captions.

## 3. The Proposed Method

To achieve unified human and animal generation, we present UNiMC, a framework that generates keypoint-guided human and animal images within a framework. We introduce the preliminaries in Apx. C.1 and the problem setting in Sec. 3.1. Then, we introduce UNiMC in Sec. 3.2, which consists of *unified keypoint encoder* and *timestep-aware keypoint modulator*. The former is a lightweight module that encodes keypoints from different species into a unified representation space, addressing semantic and structural differences across classes. The latter models keypoint and backbone tokens using self-attention to enable keypoint-

level control.

## 3.1. Problem Setting

We aim to achieve multi-class keypoint-level control in image generation using three conditioning inputs: keypoint position $\mathbf{p}$, class name $\mathbf{n}$, and bounding box $\mathbf{b}$. More formally, we aim to learn an image generation model $\mathcal{F}(\mathbf{c}_g, \{(\mathbf{n}_1, \mathbf{p}_1, \mathbf{b}_1), \ldots, (\mathbf{n}_n, \mathbf{p}_n, \mathbf{b}_n)\})$ conditioned on a global text prompt $\mathbf{c}_g$ and per-instance conditions $(\mathbf{n}_i, \mathbf{p}_i, \mathbf{b}_i)$, to generate instances at corresponding positions with various keypoints and classes.

## 3.2. UNIMC Framework

### 3.2.1. UNIFIED KEYPOINT ENCODER

Unlike the prevalent approach (Zhang et al., 2023; Ju et al., 2023b) of using skeleton images as conditions, we utilize more compact and informative condition signals. Our conditions are the keypoint coordinates, bounding box coordinates, and class names of each instance. This compact condition allows class- and instance-aware unified encoding and avoids the class binding confusion and instance binding confusion issues that are common in previous approaches. Specifically, each instance's keypoints $\mathbf{p}_i$ are expressed as a sequence of tuples $[(x_1, y_1, v_1), \ldots, (x_k, y_k, v_k)]$, where $(x_k, y_k)$ denotes the 2D coordinates and $v_k$ represents the visibility of the $k$-th keypoint. We parameterize bounding boxes by their top-left and bottom-right corners.

We convert the 2D point coordinates for each instance's keypoints $\mathbf{p}_i$ and bounding boxes $\mathbf{b}_i$ using a Fourier mapping $\gamma(\cdot)$ (Mildenhall et al., 2021). Additionally, we encode the class name $\mathbf{n}_i$ using a T5 (Chung et al., 2024) text encoder and PIXART-$\alpha$'s pretrained text projector, obtaining the class embedding $\mathbf{en}_i$. Finally, we concatenate the Fourier embedding and $\mathbf{en}_i$ and feed them to an MLP to obtain the corresponding keypoints or bounding boxes token $\mathbf{g}_i$ for the instance $i$: $\mathbf{g}_i = \mathrm{MLP}([\mathbf{en}_i, \gamma(\mathbf{p}_i)])$. We use different MLPs for keypoints and bounding boxes. Thus, for each instance $i$, we obtain $\mathbf{g}_i^{\mathrm{kpt}}$ and $\mathbf{g}_i^{\mathrm{box}}$. If an instance has only one condition, such as keypoint or bounding box, or if some points are not visible, we use a learnable mask token (He et al., 2022) $\mathbf{e}_i$ to represent the invisible parts:

$$\mathbf{g}_i = \mathrm{MLP}([\mathbf{en}_i, \ s \cdot \gamma(\mathbf{p}_i) + (1 - s) \cdot \mathbf{e}_i]) \qquad (1)$$

where $s$ is a binary vector indicating the presence of a specific condition's part. Finally, we reorder and concatenate the tokens to group each instance's keypoint token $\mathbf{g}_i^{\mathrm{kpt}}$ with its bounding box token $\mathbf{g}_i^{\mathrm{box}}$, forming a combined token $\mathbf{g}_i^{\mathrm{all}}$. This process makes the model class-aware, enabling it to encode keypoints from different classes into the shared space.

### 3.2.2. TIMESTEP-AWARE KEYPOINT MODULATOR

As shown in Fig. 3 *Left*, we insert self-attention layers into the blocks of PIXART-$\alpha$ to model the relationship between

unified keypoint feature tokens and backbone feature tokens, enabling keypoint-level control. Fig. 3 *Right* presents the modulator under different configurations, where $g$ is a summation function. Configuration (a) works best, named ***timestep-aware keypoint modulator***.

Diffusion models exhibit different spatial structural features at different timesteps (Hertz et al., 2022; Tumanyan et al., 2023). To better utilize keypoint features, we propose an **all blocks shared** timestep adapter:

$$E_s = (\gamma_{share}, \beta_{share}) = \mathcal{T}_{adapter}(\mathbf{t})$$

This adapter shares the structure with PIXART-$\alpha$'s *AdaLN-single* module, enabling the module to be aware of both the structure and the timestep condition. Each added module's features are modulated by the scale $\gamma_{share}$ and shift $\beta_{share}$.

Specifically, we denote the $m$ tokens from the backbone as $\mathbf{V}$. We concatenate the tokens $\mathbf{g}_i^{\mathrm{all}}$ for $n$ instances into $\mathbf{G}$. We apply Layer Norm (LN) (Ba et al., 2016) and Multi-Head Self-Attention (MHSA) to $\mathbf{G}$ and $\mathbf{V}$, then select the first $m$ tokens:

$$\widetilde{\mathbf{V}} = \mathrm{MHSA}(LN([\mathbf{V}, \mathbf{G}])[: m] \qquad (2)$$

Each *timestep-aware keypoint modulator* incorporates block-wise learnable scale and shift $E_i = (\gamma_i, \beta_i)$, where $i$ represents the layer index within the backbone. The following operations are performed in this layer:

$$\widetilde{\mathbf{V}} = (\gamma_i + \gamma_{share})\widetilde{\mathbf{V}} + (\beta_i + \beta_{share}) \qquad (3)$$

To preserve the model's initial capabilities, $\widetilde{\mathbf{V}}$ is passed through a zero-initialized linear layer and then added as a residual to $\mathbf{V}$. We denote the structure described above as **Config a**, and we also explore three other structures:

**Config b: Naive Gated SA.** We experiment with using the Naive Gated SA architecture directly from GLIGEN (Li et al., 2023). The structure is illustrated in Fig. 3 (b). The learnable scalars $\gamma_a$ and $\gamma_d$ are initialized to zero.

**Config c: Time Embedding Using Pretrained.** Building on **Config a**, we explore using the output of the first layer of the pretrained time encoder from PIXART-$\alpha$ to replace $\gamma_{share}$ and $\beta_{share}$, denoting it as $E_p$. The structure is illustrated in Fig. 3 (c). The aim is to explore whether the pretrained timestep features are effective in *timestep-aware keypoint modulator*.

**Config d: Block-Wise adaLN-Zero Time Embedding.** Building on **Config a**, we omit $\mathcal{T}_{adapter}$ and the block-wise learnable parameters $\gamma_i$ and $\beta_i$. Instead, we use *adaLN-Zero* for each block as in DiT (Peebles & Xie, 2023). The time encoder for the $i$-th layer is denoted as $\mathcal{T}_i$, and its output is $\mathcal{T}_i(t)$. The structure of this module is illustrated in Fig. 3 (d).

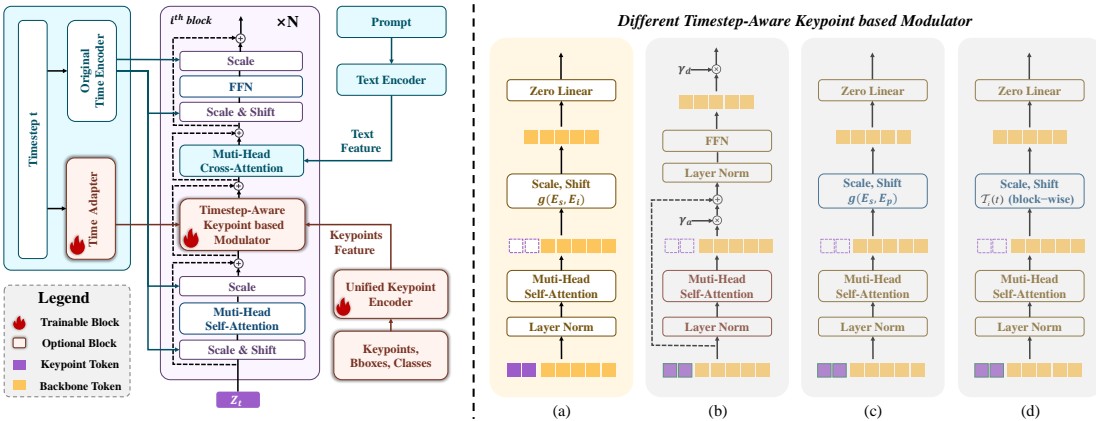

*Figure 3.* **Timestep-Aware Keypoint based Modulator.** *Left:* The block of PIXART-$\alpha$. *Right:* Four modulator variants are proposed to control DiT (detailed in Sec. 3.2.2). All variants leverage self-attention to model backbone and keypoint tokens: (a) and (c) use global-wise timestep modulation (w/o and w pretrained timestep encoder, respectively), (b) adopts GLIGEN-style gated self-attention (Li et al., 2023), and (d) applies block-wise timestep modulation. **(a) works best.**

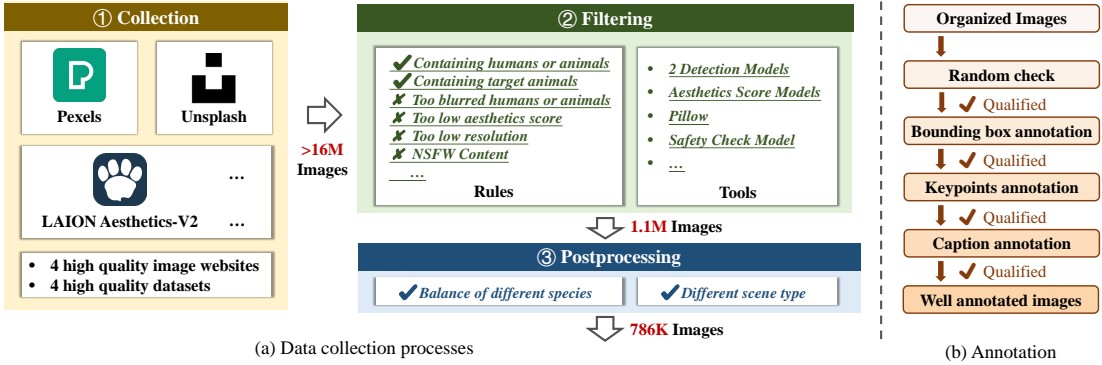

(a) Data collection processes        (b) Annotation

*Figure 4.* **Dataset Collection and Annotation Pipeline. (a)** We source over 16M images from high-quality datasets and websites. After rigorous filtering and processing, we retain 786K images. **(b)** Our data is annotated by multiple expert models and subjected to strict manual review.

## 4. The Proposed `HAIG-2.9M` Dataset

In this section, we describe the data collection, filtering, and annotation process, illustrated in Fig. 4, to construct `HAIG-2.9M`. We collect 31 non-rigid classes, including humans and 30 animal classes, from APT36K (Yang et al., 2022), we refer to these 31 non-rigid classes as our target classes. Each category has 17 corresponding keypoints. All annotation procedures involve a 5% random human check and corresponding adjustments to the annotation methods.

### 4.1. Data Collection and Filtering

To ensure data quality and scale, we crawl 460K images from four high-quality image websites: Pexels (Pexels, 2024), Pixabay (Pixabay, 2024), Stocksnap (stocksnap, 2024), and Unsplash (unsplash, 2024). Specifically, we search and crawl images using class names as tags, and rigorously check to ensure all downloaded images are non-commercial. Additionally, we filter images from four large-scale, high-quality image datasets: JourneyDB (Sun et al., 2024), SA1B (Kirillov et al., 2023), LAION-PoP (Schuhmann et al., 2024b), and LAION-Aesthetics-V2 (Schuh-

mann et al., 2024a). The four datasets originally contain over 16M images. By filtering captions to include our desired classes, we retain 2.07M images.

Then, we need to filter out images that do not contain the target classes. Specifically, we use two lightweight open-vocabulary object detection models, Grounding-Dino (Liu et al., 2023) and YOLO-World (Cheng et al., 2024), to check if the images contain the desired classes. Only when both models detect the target class do we retain the image. Additionally, we exclude images with a resolution smaller than $512 \times 512$ or an aesthetic score (christophschuhmann, 2023) below 5.0. We also remove images with overly blurred humans or animals, detected using a combination of edge detection (Canny, 1986) and blurriness metrics based on the Laplacian variance method (Pertuz et al., 2013). Furthermore, all images undergo a safety checker (CompVis, 2024) to filter NSFW (Laborde, 2024) content. We ensure balance in the number of instances across different classes by dropping classes with excessive quantities. Ultimately, we obtain 786K images after filtering, with 82.6K filtered from

web crawling and 703.4K filtered from existing datasets.

## 4.2. Data Annotation

To ensure high-quality annotations on such a large-scale dataset, we randomly sample 5K images. For each annotation type, we select multiple expert models to annotate the chosen data. We then conduct a user study involving well-trained 10 users to compare the outputs of different models, allowing them to choose the one with the best annotation quality. The model with the highest score is subsequently used to annotate the entire dataset.

**Bounding Box Annotation.** We compare five expert models: YOLO-World (Cheng et al., 2024), Grounding-DINO (Liu et al., 2023), DITO (Kim et al., 2024), OWL-ViT (Minderer et al., 2024), and CoDet (Ma et al., 2024a). Ultimately, we select YOLO-World as our annotation model, as shown in the voting results in Apx. A.3 (a).

**Human Keypoint Annotation.** We evaluate three expert models: DWPose (Yang et al., 2023), RTMPose (Jiang et al., 2023), and OpenPose (Cao et al., 2017). We choose DWPose as our annotation model, with the voting results presented in Apx. A.3 (b).

**Animal Keypoint Annotation.** We assess three expert models: ViTPose++H (Xu et al., 2022), X-Pose (Yang et al., 2025), and SuperAnimals (Ye et al., 2024). ViTPose++H is selected as our annotation model, and the voting results can be found in Apx. A.3 (c).

**Caption Annotation.** We compare four expert models: GPT4o (OpenAI, 2024b), CogVLM2 (Wang et al., 2023), LLaVA-1.6-34B (Liu et al., 2024a), and Qwen-VL (Bai et al., 2023). We choose GPT4o and CogVLM2 as our annotation models, using 1:4 ratio for the number of samples annotated. The voting results are shown in Apx. A.3 (d).

The model comparisons and manual checks of annotation results involved approximately 200 person-hours.

## 4.3. Data Statistics and Visualization

**Statistics Overview.** Tab. 1 provides a comparative analysis of mainstream human and animal keypoint-image paired datasets and HAIG-2.9M, as well as comparisons with mainstream high-quality T2I datasets. Unlike typical datasets such as COCO (Lin et al., 2014), Human-Art (Ju et al., 2023a), and APT36K (Yang et al., 2022), which are exclusively annotated for either humans or animals , our dataset, HAIG-2.9M, uniquely annotates both. Moreover, HAIG-2.9M covers a more diverse range of scene types and significantly surpasses previous datasets in terms of the number of images and instances, with over ten times the number of images and instances compared to these datasets. Additionally, our dataset features higher aesthetic scores, longer caption lengths, and higher average resolution. In

HAIG-2.9M, 22.6% of the images contain only animals, 24.6% contain both humans and animals, and 52.8% contain only humans. More statistics about HAIG-2.9M can be found in Apx. A. Fig. 5(a) illustrates the diversity of styles in HAIG-2.9M, while Fig. 5(b) displays an example of our annotation. Additionally, Fig. 1 **Top** also showcases several examples.

*Table 2.* Split of HAIG-2.9M.

| Dataset | Image | Instance | Class | Multi-Class | Multi-Instance |
|---|---|---|---|---|---|
| Training Set | 745,828 | 2,725,484 | 31 | 183,376 | 361,816 |
| Validation Set | 39,342 | 145,504 | 31 | 9,560 | 19,068 |
| Testing Set | 1,224 | 3,785 | 31 | 656 | 748 |
| Total | 786,394 | 2,874,773 | 31 | 193,592 | 381,632 |

**Dataset Split.** Detailed statistics for each subset of the dataset are provided in Tab. 2. First, for the testing set, we select 40 images for each class, ensuring that each class of images contains multiple classes. Then, we split the remaining images into training and validation sets at an approximately $20 : 1$ ratio. We adopt a class-level partition to ensure the class proportions are balanced between the training and validation sets. The training set comprises $745K$ images and $2.7M$ instances, while the validation set consists of $39K$ images and $145K$ instances.

## 5. Experiments

**Implementation Details.** We use PIXART-$\alpha$-1024px (Chen et al., 2024c) as backbone. We use the AdamW optimizer (Loshchilov & Hutter, 2017) with a weight decay of 0.03 and a fixed learning rate of $2e-5$, we only train the *unified keypoint encoder* and the *timestep-aware keypoint modulator*. We train at $1024 \times 1024$ resolution for 8K steps with a batch size of 256 using 8 A800 GPUs. During training, we drop the bounding box condition with 50% probability, the keypoint condition with 15% probability, and the prompt with 10% probability. For evaluation, we utilize the testing set of HAIG-2.9M.

**Comparison Methods.** We compare UNIMC with three categories of representative methods: **1)** Base model PIXART-$\alpha$; **2)** ControlNet, which uses skeleton images as conditions; **3)** GLIGEN, which uses keypoint coordinates as conditions.

**Comparison Datasets.** We experiment with four dataset configurations for training: **1)** HAIG-2.9M; **2)** Human dataset COCO; **3)** Animal dataset APT36K; and **4)** a combination of COCO and APT36K.

**Evaluation Metrics.** We adopt commonly-used metrics for comprehensive comparisons from five perspectives: **1)** *Image Quality*. FID (Heusel et al., 2017) and KID (Bińkowski et al., 2018) reflect quality and diversity; **2)** *Text-Image Alignment*. CLIP (Radford et al., 2021) text-image similarity is reported; **3)** *Class Accuracy*. YOLO-World detects each class and calculates the maximum IoU between detection boxes and the Ground Truth box. If the IoU is higher than 0.5, the class is considered correctly generated; **4)** *Pose*

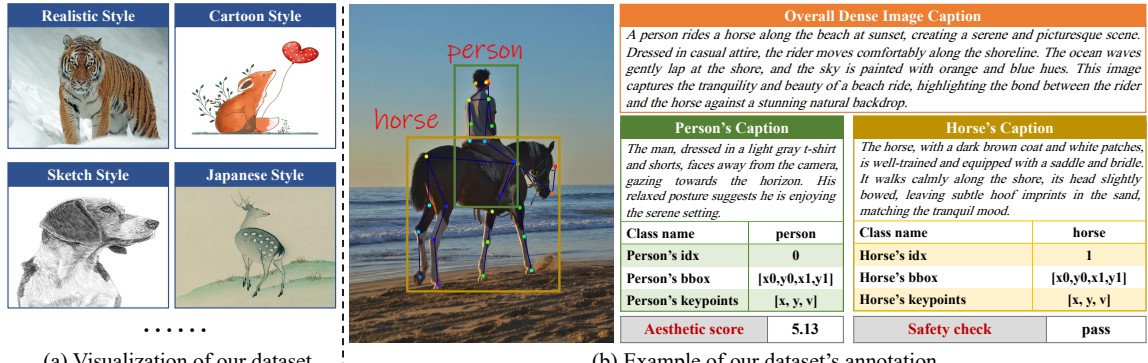

(a) Visualization of our dataset | (b) Example of our dataset's annotation

*Figure 5.* **Dataset Visualization. (a)** `HAIG-2.9M` contains images of various styles. **(b)** shows an example from our annotated dataset. *For visualization, we render the bounding box and keypoint as images overlaid on the original image.* The top right corner displays our detailed captions, including a **overall dense image caption**, **person's caption**, and **horse's caption**. The two boxes in the middle respectively show the class name, ID, bounding box, and keypoint annotations for the human and horse separately. For brevity, we omit the specific data for bounding boxes and keypoints. At the bottom, we indicate the aesthetic score and whether the image passed the safety check.

"Indoor garden scene with a **woman** gazing at a small **giraffe**, natural light streaming in."

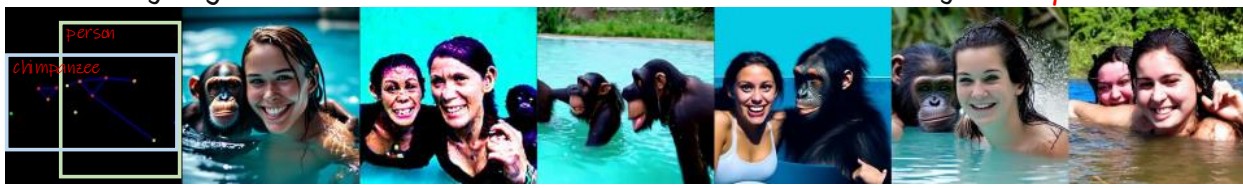

"A young, wet-haired **woman** smiles into the camera, encircled by a **chimpanzee**…"

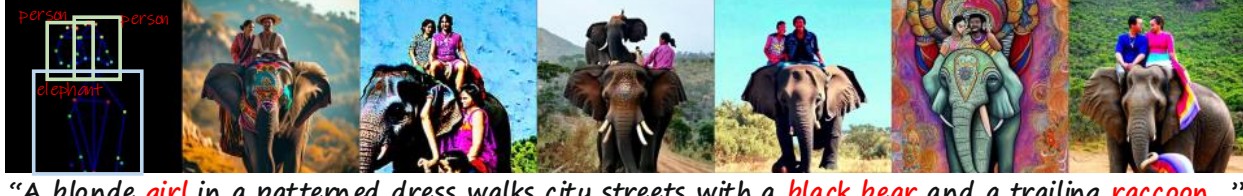

"A **man** and **woman** in traditional dress ride an elaborately painted **elephant** on a hillside…"

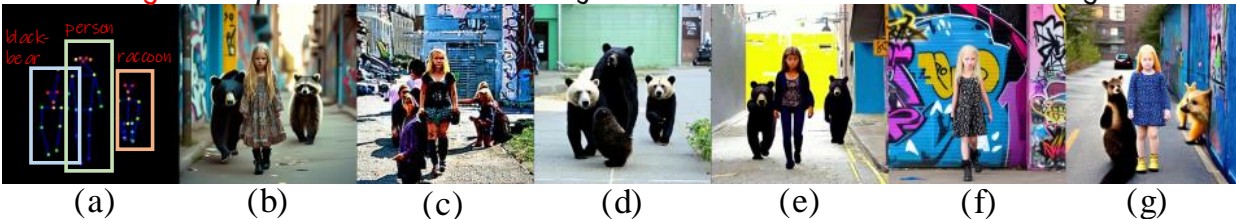

"A blonde **girl** in a patterned dress walks city streets with a **black bear** and a trailing **raccoon**…"

| (a) | (b) | (c) | (d) | (e) | (f) | (g) |

*Figure 6.* **Qualitative Comparisons.** We provide all methods with required condition formats. **From left to right: (a)**: Input condition, **(b)**: UniMC trained on `HAIG-2.9M`, **(c)**: UniMC trained on COCO, **(d)**: UniMC trained on APT36K, **(e)**: UniMC trained on COCO+APT36K, **(f)**: ControlNet, **(g)**: GLIGEN.

*Accuracy.* We use DWPose to extract human poses and ViTpose++H to extract animal poses from synthetic images and compare them with the input pose conditions. Average Precision (AP) (Lin et al., 2014) is reported. **5)** *Human*

*Subjective Evaluation.* We present the human preference study in Apx. D.3.

*Table 3.* **Quantitative Results on Different Methods**. We compare our model with PIXART-$\alpha$ (Chen et al., 2024c) and keypoint-guided methods (Zhang et al., 2023; Li et al., 2023). For PIXART-$\alpha$ and UNIMC, we first generate $1024 \times 1024$ results, then resize back to $512 \times 512$.

| Methods | Image Quality | | Alignment | Class Accuracy (%) | | Pose Accuracy (AP) | |
| --- | --- | --- | --- | --- | --- | --- | --- |
| | FID $\downarrow$ | KID$_{\times 1k}$ $\downarrow$ | CLIP $\uparrow$ | Human $\uparrow$ | Animal $\uparrow$ | Human $\uparrow$ | Animal $\uparrow$ |
| PIXART-$\alpha$ | **23.50** | **7.52** | 32.17 | 23.06 | 13.79 | 0.19 | 0.06 |
| ControlNet | 26.72 | 8.90 | 32.20 | 90.67 | 52.03 | 29.01 | 20.29 |
| GLIGEN | 31.01 | 11.15 | 31.44 | 82.06 | 9.90 | 26.58 | 3.96 |
| UNIMC (ours) | 23.63 | 7.57 | **32.28** $_{0.2\%\uparrow}$ | **93.55** $_{3.2\%\uparrow}$ | **91.71** $_{76.3\%\uparrow}$ | **30.01** $_{3.4\%\uparrow}$ | **28.38** $_{39.9\%\uparrow}$ |

*Table 4.* **Quantitative Results on Different Training Datasets.** We bold the textbfbest and underline the second results. Our improvements over the second method are shown in red.

| Dataset | Image Quality | | Alignment | Class Accuracy (%) | | Pose Accuracy (AP) | |
| --- | --- | --- | --- | --- | --- | --- | --- |
| | FID $\downarrow$ | KID$_{\times 1k}$ $\downarrow$ | CLIP $\uparrow$ | Human $\uparrow$ | Animal $\uparrow$ | Human $\uparrow$ | Animal $\uparrow$ |
| COCO | 45.08 | 18.75 | 28.07 | 45.33 | 4.63 | 19.25 | 1.52 |
| APT36K | 43.72 | 18.20 | 29.00 | 21.96 | 48.95 | 5.64 | 19.52 |
| COCO+APT36K | 33.79 | 13.55 | 32.10 | 67.97 | 67.83 | 25.69 | 21.07 |
| `HAIG-2.9M` (ours) | **23.63** $_{30.1\%\downarrow}$ | **7.57** $_{44.1\%\downarrow}$ | **32.28** $_{0.6\%\uparrow}$ | **93.55** $_{37.6\%\uparrow}$ | **91.71** $_{35.2\%\uparrow}$ | **30.01** $_{16.8\%\uparrow}$ | **28.38** $_{34.7\%\uparrow}$ |

## 5.1. Comparison with Prior Methods

**Qualitative Analysis.** As shown in Fig. 6 (a), (f) and (g), ControlNet sometimes fails to distinguish classes (e.g., row 1(f)) and struggles with accurate animal pose control (e.g., row 2 and 5(f)). GLIGEN tends to generate humans (e.g., row 2(g)) and has difficulty controlling animal poses, leading to significantly degraded image quality (e.g., rows 1 and 4(g)). In contrast, UNIMC achieves fine-grained control of poses for different classes and instances while maintaining high-quality generation, even in the presence of heavy occlusion (e.g., rows 2 and 3(b)).

**Quantitative Analysis.** As shown in Tab. 3, PIXART-$\alpha$ and UNIMC outperform ControlNet and GLIGEN in terms of image quality and text-image alignment. UNIMC significantly surpasses the baseline models in class accuracy and pose accuracy, especially for animal pose accuracy.

Both qualitative and quantitative results demonstrate that: **1)** *unified keypoint encoder* outperforms previous methods by encoding richer class and keypoint information into a compact and informative feature space. **2)** *timestep-aware keypoint modulator* effectively leverages keypoint tokens to modulate backbone, enabling fine-grained keypoint control.

## 5.2. Comparison with Different Training Dataset

**Qualitative Analysis.** As shown in Fig. 6(a)-(e), training only on COCO struggles to generate animals, and when it does, the poses are incorrect. Similarly, training only on APT struggles to control human generation. Training on COCO+APT36K can generate both humans and animals, but it often generates incorrect class (e.g., rows 4(e)) and struggles with fine-grained pose control. All three scenarios degrade generation quality significantly. In contrast, training

on `HAIG-2.9M` significantly improves generation quality and enhances class and pose control capabilities.

**Quantitative Analysis.** As shown in Tab. 4, training only on COCO results in very poor animal generation capabilities. Since most of the testing set includes animals, the model struggles with animal generation, leading to a decline in overall quality. Similarly, training only on APT36K leads to poor human generation capabilities. However, the base model's richer human priors help maintain some human generation ability. Joint training on COCO and APT36K improves the ability to generate both humans and animals but still results in low-quality generation. The model trained on `HAIG-2.9M` outperforms all other configurations across all metrics.

## 5.3. Abalation Study on Model Design

As shown in Tab. 7 of appendix, we compare the four configurations mentioned in Sec. 3 . The performance of **Config a** is noticeably superior to the others. This demonstrates that: **1)** The commonly used GLIGEN-style gated self-attention in Unet-based diffusion models is not suitable for DiT, leading to the worst performance. **2)** Global-wise timestep modulation outperforms the more parameter-heavy block-wise timestep modulation. **3)** Using a newly introduced, non-pretrained timestep encoder is essential. Additionally, we compare **the impact of inserting the controller at different layers** under Config a. By default, the controller is inserted in all 28 layers. Config a(b) involves inserting the controller in the first 14 layers, Config a(c) in the last 14 layers, and Config a(d) in both the first 7 and the last 7 layers. The results indicate that inserting the controller in all layers yields the best performance.

# 6. Conclusion

In this paper, we present UNiMC, a DiT-based framework specifically designed for high-quality keypoint-guided image generation of both humans and animals. UNiMC is capable of generating realistic, multi-class, and multi-instance images, leveraging keypoint-level, instance-level, and dense semantic conditions to achieve precise and controllable generation of non-rigid objects. Additionally, we introduce `HAIG-2.9M`, the first large-scale, high-quality, and diverse dataset annotated with keypoint-level, instance-level, and densely semantic labels for both humans and animals, designed for keypoint-guided human and animal image generation. We will discuss limitations in Apx. E.

# Impact Statement

This paper presents work whose goal is to advance the field of Machine Learning. Our model enhances creativity with precise keypoint-level control but carries misuse risks, we emphasize responsible use and transparency.

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

## A. Dataset

### A.1. Licenses

Image Websites:

- Pexels[1] (Pexels, 2024): Creative Commons CC0 license.

- Pixabay[2] (Pixabay, 2024): Creative Commons CC0 license.

- Stocksnap[3] (stocksnap, 2024): Creative Commons CC0 license.

- Unsplash[4] (unsplash, 2024): Unsplash+ License[5].

Image Datasets:

- JourneyDB[6] (Sun et al., 2024): JourneyDB customized license[7].

- SA-1B[8] (Kirillov et al., 2023): SA-1B Dataset Research License[9].

- LAION-PoP[10] (Schuhmann et al., 2024b): Creative Common CC-BY 4.0 license.

- LAION-Aesthetics-V2[11] (Schuhmann et al., 2024a): Creative Common CC-BY 4.0 license.

Our License: Creative Common CC-BY 4.0 license.

### A.2. Statistics

The number of instances per class in HAIG-2.9M is shown in Tab. 6. Notably, because it is difficult to obtain a large number of images for "black bear" and "polar bear", we use "bear" as a supplement to the 31 classes. Due to the scarcity of some classes (e.g., rhino, hippo, and buffalo), our data has a long-tail distribution problem. However, unlike perception tasks that require a balanced number of classes, we can effectively control the poses of classes with few training samples by leveraging the priors of large-scale T2I models (e.g., gorilla in Fig. 1 bottom, chimpanzee in Fig. 6 row 4, cheetah in Fig. 6 row 5, buffalo in Fig. 7 row 6, rhino in Fig. 8 row 1 and 2, and hippo in Fig. 8 row 3).

### A.3. Annotation Comparison

We present the model comparison voting results mentioned in Sec. 4.2 in Apx. A.3, where the **Votes** column represents the total number of votes.

## B. Related Works

**Diffusion Transformer.** The Transformer architecture (Vaswani et al., 2017) has achieved success in various domains. In diffusion models, DiT (Peebles & Xie, 2023) and UViT (Bao et al., 2023) pioneer the Transformer-based diffusion model, with subsequent works refining the architecture (Hatamizadeh et al., 2023; Ma et al., 2024b; Lu et al., 2024; Tian et al., 2024) or improving training efficiency (Gao et al., 2023; Zheng et al., 2024). For Text-to-Image (T2I) synthesis, the

---

[1] https://www.pexels.com/
[2] https://pixabay.com/
[3] https://stocksnap.io/
[4] https://unsplash.com/
[5] https://unsplash.com/license
[6] https://journeydb.github.io/
[7] https://journeydb.github.io/assets/Terms_of_Usage.html
[8] https://ai.meta.com/datasets/segment-anything/
[9] https://ai.meta.com/datasets/segment-anything/
[10] https://laion.ai/blog/laion-pop/
[11] https://laion.ai/blog/laion-aesthetics/

*Table 5.* Voting Results for Model Selection

(a) Bounding Box Annotation

| Model | Votes |
|---|---|
| YOLO-World | 26199 |
| Grounding-DINO | 3469 |
| DITO | 8055 |
| OWL-ViT | 4574 |
| CoDet | 7703 |

(b) Human Keypoint Annotation

| Model | Votes |
|---|---|
| DWPose | 36709 |
| RTMPose | 10271 |
| OpenPose | 3020 |

(c) Animal Keypoint Annotation

| Model | Votes |
|---|---|
| ViTPose++H | 32092 |
| X-Pose | 11089 |
| SuperAnimals | 6819 |

(d) Caption Annotation

| Model | Votes |
|---|---|
| GPT4o | 18875 |
| CogVLM2 | 18270 |
| LLaVA-1.6-34B | 4096 |
| Qwen-VL | 8759 |

*Table 6.* **Class Counts in the `HAIG-2.9M`, Sorted Alphabetically**

| Class | Count | Class | Count | Class | Count | Class | Count |
|---|---|---|---|---|---|---|---|
| Antelope | 2755 | Deer | 20701 | Monkey | 5658 | Pig | 4018 |
| Bear | 22833 | Dog | 114502 | Orangutan | 126 | Polar Bear | 5625 |
| Black Bear | 304 | Elephant | 49457 | Panda | 1521 | Rabbit | 623 |
| Buffalo | 1219 | Fox | 7536 | Person | 2159681 | Raccoon | 1962 |
| Cat | 36396 | Giraffe | 8508 | Pig | 4018 | Rhino | 178 |
| Cheetah | 2249 | Gorilla | 513 | Sheep | 94183 | Tiger | 4344 |
| Chimpanzee | 321 | Hippo | 1080 | Howling Monkey | 1721 | Wolf | 8976 |
| Cow | 100689 | Horse | 205616 | Lion | 4400 | Zebra | 5444 |

PIXART series (Chen et al., 2024c;b;a) demonstrated more efficient training and higher quality image generation compared to UNet-based models (Rombach et al., 2022; Podell et al., 2024). Other works (Esser et al., 2024; Chen et al., 2023; OpenAI, 2024a; Gao et al., 2024; Li et al., 2024b; Xie et al., 2023; Nair et al., 2024) have also proven the efficiency, potential and scalability of DiT. In this work, we use PIXART-$\alpha$ as our backbone, which is a variant of DiT (Peebles & Xie, 2023). We explore various DiT control variants to achieve unified keypoint-level control.

## C. Method

### C.1. Preliminaries

**Diffusion Probabilistic Models.** Diffusion models (Ho et al., 2020; Sohl-Dickstein et al., 2015) define a forward diffusion process to gradually convert the sample $\mathbf{x}$ from a real data distribution $p_{\text{data}}(\mathbf{x})$ into a noisy distribution, and learn the reverse process in an iterative denoising way (Sohl-Dickstein et al., 2015; Ho et al., 2020). During the sampling process, the model can transform Gaussian noise of normal distribution to real samples step-by-step. The denoising network $\epsilon_{\boldsymbol{\theta}}(\cdot)$ estimates the additive Gaussian noise, which is typically structured as a DiT (Peebles & Xie, 2023) or a UNet (Ronneberger et al., 2015) to minimize the mean-squared error:

$$\min_{\boldsymbol{\theta}} \ \mathbb{E}_{\mathbf{x},\mathbf{c},\epsilon,t} \ \left[||\epsilon - \epsilon_{\boldsymbol{\theta}}(\sqrt{\hat{\alpha}_t}\mathbf{x} + \sqrt{1 - \hat{\alpha}_t}\epsilon, \mathbf{c}, t)||_2^2\right], \tag{4}$$

where $\mathbf{x}, \mathbf{c} \sim p_{\text{data}}$ are the sample-condition pairs from the training distribution; $\epsilon \sim \mathcal{N}(\mathbf{0}, \mathbf{I})$ is the ground-truth noise; $t \sim \mathcal{U}[1, T]$ is the time-step and $T$ is the predefined diffusion steps; $\hat{\alpha}_t$ is the coefficient decided by the noise scheduler.

**Latent Diffusion Model & PIXART-$\alpha$.** The widely-used latent diffusion model (LDM) (Rombach et al., 2022), performs the denoising process in a separate latent space to reduce the computational cost. Specifically, a pre-trained VAE (Esser et al., 2021) first encodes the image $\mathbf{x}$ to latent space $\mathbf{z} = \mathcal{E}(\mathbf{x})$ for training. At the inference stage, we can reconstruct the generated image through the decoder $\hat{\mathbf{x}} = \mathcal{D}(\hat{\mathbf{z}})$. In this work, we use PIXART-$\alpha$ (Chen et al., 2024c) as our backbone, which is a basic T2I model based on the Latent Diffusion Transformer. It achieves higher generation quality while the model parameters and training data are much smaller than those of the UNet-based SD series models (Rombach et al., 2022; Podell et al., 2024).

# D. More Results

## D.1. Ablation Study of Different Configs of Timestep-aware Keypoint based Modulator

As shown in Tab. 7, we compare the four configurations mentioned in Sec. 3, from **Config a** to **Config d**. The performance of Config a is noticeably superior to the others. This demonstrates that: **1)** The commonly used GLIGEN-style gated self-attention in Unet-based diffusion models is not suitable for the DiT architecture, leading to the worst performance. **2)** Global-wise timestep modulation outperforms the more parameter-heavy block-wise timestep modulation. **3)** Using a newly introduced, non-pretrained timestep encoder is essential.

Additionally, we compare **the impact of inserting the controller at different layers** under Config a. By default, the controller is inserted in all 28 layers. Config a(b) involves inserting the controller in the first 14 layers, Config a(c) in the last 14 layers, and Config a(d) in both the first 7 and the last 7 layers. The results indicate that inserting the controller in all layers yields the best performance.

*Table 7.* **Ablation on Different Configs of Timestep-Aware Keypoint based Modulator.**

| Methods | Image Quality | | Alignment | Class Accuracy (%) | | Pose Accuracy (AP) | |
| --- | --- | --- | --- | --- | --- | --- | --- |
| | FID $\downarrow$ | KID$_{\times 1k}$ $\downarrow$ | CLIP $\uparrow$ | Human $\uparrow$ | Animal $\uparrow$ | Human $\uparrow$ | Animal $\uparrow$ |
| Config b | 42.97 | 15.60 | 28.02 | 51.97 | 53.44 | 21.10 | 18.05 |
| Config c | 38.66 | 13.58 | 29.30 | 56.92 | 54.20 | 21.99 | 19.26 |
| Config d | 33.90 | 12.70 | 30.39 | 62.45 | 64.37 | 24.38 | 23.55 |
| Config a(b) | 27.93 | 10.04 | 30.12 | 75.33 | 75.01 | 28.09 | 26.55 |
| Config a(c) | 26.00 | 9.28 | 31.29 | 82.40 | 79.85 | 28.81 | 27.95 |
| Config a(d) | 27.09 | 9.62 | 31.81 | 80.07 | 77.40 | 28.46 | 27.28 |
| Config a | **23.63** $_{9.1\%\downarrow}$ | **7.57** $_{18.4\%\downarrow}$ | **32.28** $_{1.5\%\uparrow}$ | **93.55** $_{13.5\%\uparrow}$ | **91.71** $_{14.9\%\uparrow}$ | **30.01** $_{4.2\%\uparrow}$ | **28.38** $_{1.5\%\uparrow}$ |

## D.2. More Qualitative Comparisons

More qualitative comparisons shown in Fig. 7 and Fig. 8.

## D.3. Human Preference Study

The human preference study is a crucial subjective metric for evaluating the quality of generative models. We present the user study results from two aspects: **1) generation quality** and **2) control ability**. We randomly selected 10 samples from the test set and asked 10 participants to evaluate them from two perspectives. For generation quality, participants assessed which method produced the highest quality images. For class and keypoint control ability, they selected the method that generated images where the class and corresponding keypoints most accurately matched the given conditions. We then recorded the number of participants who chose each option. Tab. 8 presents the user study results across different methods, and Tab. 9 shows the user study results across different training datasets.

*Table 8.* Comparison of human evaluations across different methods, supplementing Table 4 in the main paper.

| Methods | Generation Quality | Class and Keypoint Control Ability |
| --- | --- | --- |
| Pixart | 47 | 0 |
| ControlNet | 11 | 9 |
| GLIGEN | 2 | 4 |
| UniHA (ours) | 40 | 87 |

# E. Limitations

Similar to AP10K (Yu et al., 2021), our dataset also faces a long-tail distribution problem. Additionally, our dataset has a limited number of categories, and the controllable generation of arbitrary non-rigid objects remains a challenging task.

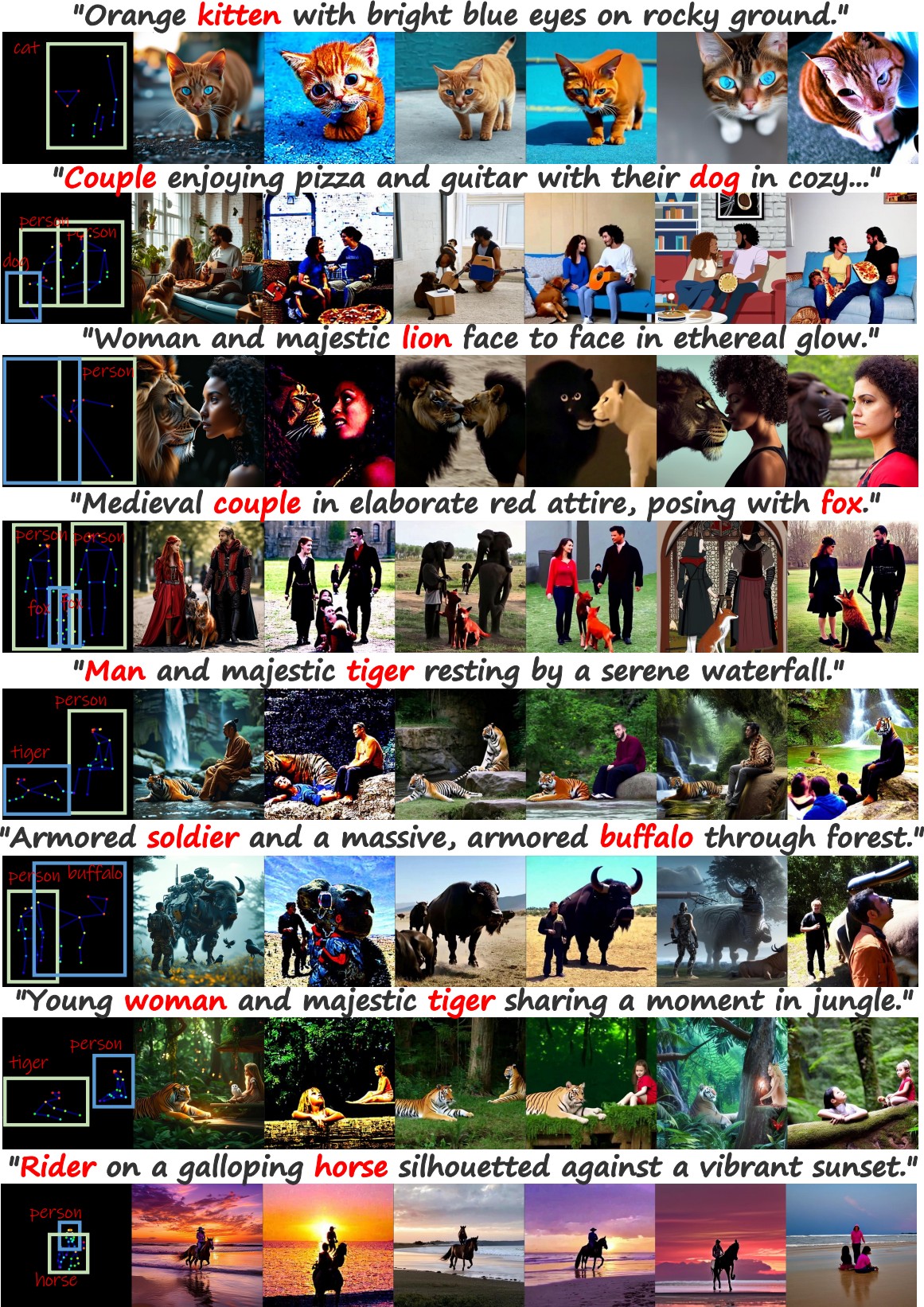

*Figure 7.* **Additional Qualitative Comparisons (I). From left to right: (a)**: Input condition, **(b)**: UNIMC trained on HAIG-2.9M, **(c)**: UNIMC trained on COCO, **(d)**: UNIMC trained on APT36K, **(e)**: UNIMC trained on COCO+APT36K, **(f)**: ControlNet, **(g)**: GLIGEN.

*Figure 8.* **Additional Qualitative Comparisons (II). From left to right: (a)**: Input condition, **(b)**: UNIMC trained on HAIG-2.9M, **(c)**: UNIMC trained on COCO, **(d)**: UNIMC trained on APT36K, **(e)**: UNIMC trained on COCO+APT36K, **(f)**: ControlNet, **(g)**: GLIGEN.

*Table 9.* Comparison of human evaluations for our method using different training datasets, supplementing Table 3 in the main paper.

| Dataset | Generation Quality | Class and Keypoint Control Ability |
|---------|--------------------|-----------------------------------|
| COCO | 2 | 0 |
| APT36K | 1 | 2 |
| COCO+APT36K | 4 | 2 |
| HAIG-2.9M (ours) | 93 | 96 |

## F. Significance of Including Animals for Conditional T2I Generation

The inclusion of animals is crucial for enhancing the diversity and generalization capabilities of conditional text-to-image generation models. Animals, like humans, are a significant part of the biological world and can be easily represented using keypoints. Animals are non-rigid and their deformations can be controlled with keypoints. They introduce a broader range of pose variations, textures, and anatomical structures, which can greatly benefit models that need to generalize across different classes. This contributes to the development of more controllable foundational visual generation models.

