# OpenReview forum: "UniMC: Taming Diffusion Transformer for Unified Keypoint-Guided Multi-Class Image Generation"
_ICML.cc/2025/Conference — ICML 2025 poster_

### Official Review · Reviewer_u8PW · 2025-02-18

**Overall Recommendation:** 4

**Summary:**

This paper proposes a dataset of human and animal images, and their keypoints, bounding boxes, and fine-grained captions. The dataset includes 786K images with 2.9M instances, averaging 3.66 instances per image. The annotations are obtained from the best among several candidates, different annotations having different best choices. Based on this dataset, the paper also proposes a controllable DiT-based framework for keypoint-guided image generation. The model can generate multiple subjects using their keypoints, class, bounding box, and a global text prompt. Both qualitative and quantitative results shows that the model significantly outperforms the previous work.

## update after rebuttal
The rebuttal addresses my most concerns. Since I already gave accept, I will keep my rating.

**Claims And Evidence:**

The dataset is for generation purpose, and thus the average aesthetic score is higher than other keypoint datasets, such as COCO.

The proposed model is tested and ablated, showing better performance than baselines.

**Essential References Not Discussed:**

N/A

**Experimental Designs Or Analyses:**

1. From Table 3, the class and pose accuracy is significantly improved for animals, but those metrics for human have only  mild improvement compared with ControlNet. What might be the reason? Can it be visualized by some attention maps or feature maps?
2. Some of the magic number should be explained. For example, why use 50% dropout for bbounding box while only 15% for keypoints? I understand it must be a performance choice, but is there any insight why? How does the dropout ratio affect the quantitative and qualitative results?

**Methods And Evaluation Criteria:**

1. This paper directly use keypoint and bounding box coordinates as controlling signal without converting them into spatial representations, such as heatmaps, but into Fourier signals, and it works better than previous work using spatial representations. It slightly counters my intuition, but also makes sense because processing in the implicit way may benefit interation better. This setting could inspire future work on image/video mulit-object interaction generation.
2. The detailed dataset processing and comparison of different methods for different annotations can benefit the community.

**Other Comments Or Suggestions:**

N/A

**Other Strengths And Weaknesses:**

N/A

**Questions For Authors:**

N/A

**Relation To Broader Scientific Literature:**

N/A

**Theoretical Claims:**

N/A

---

> ### Author Rebuttal · Authors · 2025-03-29
>
> **Dear Reviewer u8PW**,
>
> Thank you for your review and constructive comments. During the rebuttal period, we have made every effort to address your concerns. The detailed responses are below:
>
> > Q1: Why human shows mild improvement compared with ControlNet?
> >
>
> While our method shows only mild improvement on the *human* class compared to ControlNet, it brings **substantial gains across other categories**, especially animals, as shown in Table 3 of the main paper. This highlights the strength of our approach in handling diverse object classes.
>
> Our core objective is to solve the task of **keypoint-conditioned multi-class image generation**, where *human* is just one of many categories. In contrast, ControlNet is specifically designed for **human-centric** keypoint-conditioned generation, and thus naturally excels in that narrow domain.
>
> This difference is analogous to a **specialist vs. generalist** comparison. A specialist may outperform others on a single task, but a well-designed generalist can achieve strong performance across a broader spectrum. UNIMC is purposefully built as a generalist model that scales across multiple categories, not just humans.
>
> > Q2: Explanation of dropout ratio choices.
> >
>
> In our initial experiments, we applied a **15% dropout ratio** to both bounding boxes and keypoints. However, we observed that the model quickly overfit to the bounding box information—likely because **coarse object localization is much easier to learn than fine-grained keypoint patterns**. As a result, the model tended to ignore the keypoint inputs, and the pose accuracy plateaued below 20%, regardless of the number of training iterations.
>
> To address this issue, we increased the dropout ratio for bounding boxes, forcing the model to rely more on the keypoints. The table below shows how different dropout ratios for bounding boxes affect pose accuracy:
>
> | Bounding box dropout ratio | Human AP**⬆️** | Animal AP**⬆️** |
> | --- | --- | --- |
> | 0% | 14.70 | 13.92 |
> | 15% | 17.85 | 18.21 |
> | 50% | 30.01 | 28.38 |
> | 100% | 26.42 | 26.09 |
>
> These results suggest that a **moderate dropout (around 50%)** strikes a good balance, encouraging the model to make better use of keypoint information without completely discarding spatial cues from bounding boxes.

---

### Official Review · Reviewer_Rf97 · 2025-03-13

**Overall Recommendation:** 3

**Summary:**

The paper proposes a DiT based framework UniMC for keypoint guided multi-instance image generation and introduces HAIG-2.9M dataset designed for keypoint-guided human and animal image generation. Experiments are conducted on COCO, APT36K, and HAIG-2.9M datasets to show the efficacy of the proposed method.

**Claims And Evidence:**

The paper motivated the absence of a joint Human-animal keypoint datasets for training a unified model and introduced the HAIG-2.9M dataset. However, the need for a joint dataset is not justified. An ablation study on the performance for a subset of testing images that contains only multiple instances per image in HAIG-2.9M is not shown.

**Essential References Not Discussed:**

[1] Naoto Inoue, Kotaro Kikuchi, Edgar Simo-Serra, Mayu Otani, Kota Yamaguchi LayoutDM: Discrete Diffusion Model for Controllable Layout Generation, CVPR 2023.

[2] Yilin Wang, Zeyuan Chen, Liangjun Zhong, Zheng Ding, Zhuowen Tu, Dolfin: Diffusion Layout Transformers without Autoencoder, ECCV 2024

[3] Runyang Feng, Yixing Gao, Tze Ho Elden Tse, Xueqing Ma, Hyung Jin Chang DiffPose: SpatioTemporal Diffusion Model for Video-Based Human Pose Estimation, ICCV 2023

[4] Shoufa Chen, Peize Sun, Yibing Song, and Ping Luo. Diffusiondet: Diffusion model for object detection, ICCV 2023.

**Experimental Designs Or Analyses:**

Missing comparisons on related prior works such as

[1] Dewei Zhou, You Li, Fan Ma, Xiaoting Zhang, Yi Yang. MIGC: Multi-Instance Generation Controller for Text-to-Image Synthesis, CVPR 2024

[2] Xudong Wang, Trevor Darrell, Sai Saketh Rambhatla, Rohit Girdhar, Ishan Misra. InstanceDiffusion: Instance-level Control for Image Generation, CVPR, 2024

How does the proposed method compare qualitatively and quantitatively against [1,2]?

**Methods And Evaluation Criteria:**

Table 4 shows the results of using different datasets for training and evaluating on the HAIG-2.9M testing dataset. However, this is not a fair comparison since it is essentially comparing cross dataset generalization to in dataset generalization. It should also show evaluation on COCO+APT testing datasets.

**Other Comments Or Suggestions:**

The major concern is the missing comparisons with closely related prior work [1,2].

**Other Strengths And Weaknesses:**

The paper uses DiffPose for annotating the HAIG-2.9M dataset and also evaluate using the same model. This may introduce a bias in the evaluation.

**Questions For Authors:**

What is the number of images used for FID comparison?

What is the performance of HAIG with and without multiple instances in the dataset? It will be better to show the performance improvement coming from multi-instance images.

Why not annotate with the ensemble of models instead of the best model (Sec 4.2, L275).

**Relation To Broader Scientific Literature:**

The proposed method is not entirely novel as Diffusion Transformers have shown to be useful for coordinate regression in prior works [1-4]

**Theoretical Claims:**

There are no theoretical claims in the paper.

---

> ### Author Rebuttal · Authors · 2025-03-30
>
> **Dear Reviewer Rf97,**
>
> Thank you for your review and constructive comments. During the rebuttal period, we have made every effort to address your concerns. The detailed responses are below:
>
> > Q1: The need for a joint human-animal keypoint-annotated dataset.
> >
>
> As shown in **Table 4** and **Figure 6** of the main paper, training on separate human and animal keypoint datasets makes it difficult to generate images with multiple classes, limiting the development of more general keypoint-conditioned generation.
>
> > Q2: Results on multi-instance subset.
> >
>
> We present results on the **multi-instance subset** below. Our method still performs well, while baseline models decline, demonstrating that **UNIMC** performs better in multi-class scenarios.
>
> | Methods | FID ⬇️ | KID ×1k ⬇️ | CLIP **⬆️** | Human Class **⬆️** | Animal Class **⬆️** | Human AP **⬆️** | Animal AP **⬆️** |
> | --- | --- | --- | --- | --- | --- | --- | --- |
> | **ControlNet** | 27.03 | 9.15 | 31.96 | 90.65 | 31.51 | 26.47 | 9.86 |
> | **GLIGEN** | 32.51 | 12.05 | 31.04 | 80.19 | 6.31 | 25.44 | 1.58 |
> | **UNIMC** | **23.51** | **7.59** | **32.20** | **93.56** | **91.50** | **29.39** | **28.67** |
>
> > Q3: Results on COCO+APT testing datasets.
> >
>
> We have added results on the **COCO+APT testing set**, and similar to HAIG, our method demonstrates strong performance and surpasses all baseline methods.
>
> | Methods | FID ⬇️ | KID ×1k ⬇️ | CLIP **⬆️** | Human Class **⬆️** | Animal Class **⬆️** | Human AP **⬆️** | Animal AP **⬆️** |
> | --- | --- | --- | --- | --- | --- | --- | --- |
> | **ControlNet** | 25.00 | 9.12 | 30.02 | 90.01 | 30.09 | 26.92 | 9.50 |
> | **GLIGEN** | 32.41 | 12.09 | 29.32 | 80.10 | 5.09 | 24.49 | 0.90 |
> | **UNIMC** | **24.09** | **8.01** | **30.06** | **93.42** | **92.01** | **29.65** | **28.30** |
>
> > Q4: Performance with and without multiple instances in the dataset.
> >
>
> The training performance of HAIG with and without multiple instances shows that adding multi-instance images improves all metrics.
>
> | Variants | FID ⬇️ | KID ×1k ⬇️ | CLIP **⬆️** | Human Class **⬆️** | Animal Class **⬆️** | Human AP **⬆️** | Animal AP **⬆️** |
> | --- | --- | --- | --- | --- | --- | --- | --- |
> | **W/O multiple instances** | 25.00 | 8.06 | 31.00 | 91.04 | 89.09 | 24.44 | 22.65 |
> | **W multiple instances** | **23.63** | **7.57** | **32.28** | **93.55** | **91.71** | **30.01** | **28.38** |
>
> > Q5: Comparison with **[1]MIGC** and **[2]InstanceDiffusion**.
> >
>
> The table below presents the **quantitative comparison** with **MIGC** and **InstanceDiffusion**. **MIGC** controls via bounding boxes and classes, while **InstanceDiffusion** can control through bounding boxes, points, and classes. Compared to previous baseline methods, both of these models show significant improvements in class control accuracy. InstanceDiffusion, which adds point control, also improves keypoint control accuracy.
>
> | Methods | FID ⬇️ | KID ×1k ⬇️ | CLIP **⬆️** | Human Class **⬆️** | Animal Class **⬆️** | Human AP **⬆️** | Animal AP **⬆️** |
> | --- | --- | --- | --- | --- | --- | --- | --- |
> | **MIGC** | 23.78 | 8.06 | 31.29 | 92.06 | 88.04 | 17.04 | 15.99 |
> | **InstanceDiffusion** | 24.06 | 9.17 | 30.99 | 89.70 | 89.00 | 21.06 | 20.04 |
> | **UNIMC** | **23.63** | **7.57** | **32.28** | **93.55** | **91.71** | **30.01** | **28.38** |
>
> The **quantitative comparison** can be seen in [Fig.5](https://ibb.co/mFCJztLW).
>
> > Q6: What is the number of images used for FID comparison?
> >
>
> As mentioned in the **Implementation Details** section, we used the **HAIG-2.9M testing set** for evaluation, with **1,224** images.
>
> > Q7: Why not annotate with the ensemble of models?
> >
>
> Relevant works, such as Pixart-sigma, have shown that using a better annotation model improves the model's performance. Additionally, animal pose estimation is not as mature as human pose estimation, and only selecting the best model ensures that we can achieve usable results.
>
> > Q8: Bias in Evaluation.
> >
>
> Thank you for your insightful question. Please refer to **Q3** — our method also achieves strong performance on human-annotated datasets, which supports the **validity of our evaluation results** and the **reliability of our evaluation methodology**.
>
> > Q9: Difference from coordinate regression tasks.
> >
>
> Our task is to extend **keypoint-conditioned image generation** from a single category (human) to a **multi-class setting**. To achieve this, we use keypoint **coordinates as conditional inputs** to guide image generation. In contrast, prior works on **coordinate regression** use coordinates as **outputs**, not as conditioning inputs. While these works demonstrate that **DiT** can model coordinate distributions, they do not show that DiT can **generate images conditioned on coordinates**, which is the core challenge of our work. Therefore, the prior use of DiT for coordinate regression is only loosely related to our task and does not diminish the novelty of our proposed approach.

---

> > ### Comment · Reviewer_Rf97 · 2025-04-07
> >
> > Thanks to the authors for the rebuttal. It addressed most of my concerns. I am increasing my score to Weak Accept. I would recommend the authors to include the additional results and comparisons in the revised version.

---

> > > ### Author Response · Authors · 2025-04-08
> > >
> > > Dear Reviewer Rf97,
> > >
> > > Thank you sincerely for taking the time to review our rebuttal and for thoughtfully considering our clarifications. We are especially grateful that you found our additional experiments and comparisons helpful, and we truly appreciate your updated score.
> > >
> > > Your comments throughout the review process—particularly the suggestions regarding additional comparisons and ablation studies—have been instrumental in helping us strengthen the paper. We will make sure to include the extended results and analyses in the revised version as you recommended.
> > >
> > > If there is anything further you would like to discuss or clarify, we would be more than happy to engage further.
> > >
> > > Sincerely,
> > > The Authors

---

### Official Review · Reviewer_2vGV · 2025-03-14

**Overall Recommendation:** 4

**Summary:**

This paper proposes a framework (UNIMC) that generates images containing multiple objects (including humans and various other entities) by leveraging joint keypoints and introduces a large-scale dataset (HAIG-2.9M) to support this approach. Unlike conventional keypoint-based image control methods, UNIMC utilizes a unified keypoint encoder that encodes keypoints of various classes and instances into a shared representation space, enabling more precise multi-class and multi-instance image generation. By leveraging a large-scale dataset, the proposed method enhances the performance of keypoint-based image generation models, particularly in multi-object and occlusion-heavy scenarios, while maintaining high image quality.

**Claims And Evidence:**

• The paper claims that the proposed method outperforms existing keypoint-conditioned image generation models such as GLIGEN and ControlNet, which generate images containing multiple objects. This claim is supported by quantitative evaluations in the experimental results, where various metrics demonstrate the superiority of the proposed approach.

• In occlusion scenarios, qualitative evaluation results show that objects in the generated images maintain accurate poses by leveraging per-object keypoints.

• The proposed dataset can be effectively utilized for keypoint-based image generation and contains over ten times more images and objects compared to major existing datasets such as COCO and Human-Art, while also maintaining high quality. This is detailed in Table 1 of the paper.

**Essential References Not Discussed:**

While the paper compares various models and their generative performance, it does not provide a thorough comparison with the latest Visual-Language Model (VLM)-based image generation approaches. Incorporating both qualitative and quantitative evaluations against state-of-the-art models in recent research trends would have broadened the study’s scope and enhanced its credibility.

**Experimental Designs Or Analyses:**

The paper effectively employs performance metrics commonly used in generative models and conducts both quantitative and qualitative evaluations. By comparing various datasets and models, the experiments are well-structured and appear valid. The results demonstrate that UNIMC maintains accurate poses using per-object keypoints, even in occlusion scenarios, highlighting its robustness.

However, the study lacks an analysis of training and inference speed, which are crucial factors for practical application. Providing additional details on these aspects would enhance the clarity of the research. Furthermore, examining whether this method can be applied to other diffusion models would help establish it as a more generalized approach.

**Methods And Evaluation Criteria:**

The proposed method introduces a unified keypoint encoder trained to generate high-quality images that accurately distinguish various classes and multiple objects. Previous approaches struggled with keypoint-based control due to inconsistencies in keypoint formats across classes and challenges in differentiating objects within the same class. This method effectively addresses these issues, marking a significant improvement.

Unlike existing models that rely solely on joint keypoints for generating multi-class and multi-object images, the proposed approach takes a different direction. By integrating both keypoint and class information during training, it offers a more structured and intuitive solution. This innovation appears both practical and effective. However, a limitation of the study is that it does not verify whether the method can be applied to other models, leaving room for further validation.

**Other Comments Or Suggestions:**

• The contributions of the paper are somewhat repetitive, and it would be beneficial to highlight additional novel contributions.

**Other Strengths And Weaknesses:**

Strengths

This study introduces a new dataset designed for multi-class and multi-object image generation based on joint keypoints, addressing the traditional limitation where such datasets were primarily designed for recognition tasks. To ensure accurate annotations, even in images with overlapping objects, bounding boxes, keypoints, and captions were incorporated, ultimately enhancing the quality of image generation.

Weaknesses

While the paper demonstrates the superiority of the proposed method through comparative experiments with GLIGEN and ControlNet, it lacks a detailed analysis explaining why it outperforms existing approaches. Additionally, a comparison with Fusion Encoder-based Text-to-Image models, which align with recent research trends, would have further strengthened the study’s impact. Furthermore, while the HAIG-2.9M dataset and the UNIMC model are repeatedly emphasized as key contributions, additional novel elements would help further differentiate this work from prior research.

**Questions For Authors:**

.

**Relation To Broader Scientific Literature:**

The paper presents a detailed comparison with ControlNet and GLIGEN, effectively illustrating the limitations of existing methods. It directly evaluates the proposed model against PIXART-α, its base model, as well as other keypoint-conditioned image generation models like ControlNet and GLIGEN. However, a broader comparison with various text-based image generation models would have further strengthened the study.

**Theoretical Claims:**

N/A

---

> ### Author Rebuttal · Authors · 2025-03-30
>
> **Dear Reviewer 2vGV,**
>
> Thank you for your review and constructive comments. During the rebuttal period, we have made every effort to address your concerns. The detailed responses are provided below:
>
> > Q1: Applicability to other models.
>
> Our method is designed to be compatible with **almost all DiT-based architectures**, due to the following reasons:
>
> 1. The **Unified Keypoint Encoder** is model-agnostic and does not rely on any specific architectural design.
> 2. The **Timestep-aware Keypoint Modulator** is implemented using only self-attention and does not depend on any model-specific components.
>
> To demonstrate the general applicability of our method, we conducted experiments on [Hunyuan-DiT](https://github.com/Tencent/HunyuanDiT). The specific experimental results are shown in the table below. Our method successfully equips **Hunyuan-DiT** with the ability to perform **keypoint-conditioned multi-class image generation**.
>
> | **Methods**           | **Human Class ⬆️** | **Animal Class ⬆️** | **Human AP ⬆️** | **Animal AP ⬆️** |
> | --------------------- | ----------------- | ------------------ | -------------- | --------------- |
> | Hunyuan-DiT           | 25.49             | 16.77              | 0.26           | 0.14            |
> | **Hunyuan-DiT+UNIMC** | **94.06**         | **91.89**          | **30.92**      | **28.75**       |
>
> > Q2: Analysis of training and inference speed.
>
> Please refer to **R1-Q1** and **R1-Q4**. Our method is highly efficient in both training and inference.
>
> Specifically, it requires only **576 A800 GPU-hours** to complete training. At inference time, it is **more efficient than GLIGEN** and **comparable to ControlNet**. Moreover, our method can quickly achieve strong performance even when fine-tuned on small-scale datasets.
>
> > Q3: Comparison with various text-based and Visual-Language Model (VLM)-based image generation models.
>
> Our method is orthogonal to general text-to-image (T2I) or Visual-Language Model (VLM)-based image generation approaches. These models lack the ability to generate images conditioned on **keypoints from different object categories**. In contrast, we introduce a framework that enables **keypoint-conditioned multi-class image generation**.
>
> The table below provides a comparison between **UNIMC** and recent state-of-the-art T2I models. Notably, our method achieves **substantial improvements in Human/Animal class accuracy and keypoint-level AP**, demonstrating its superiority in controllability and structural alignment.
>
> | **Methods** | **Human Class ⬆️** | **Animal Class ⬆️** | **Human AP ⬆️** | **Animal AP ⬆️** |
> | ----------- | ----------------- | ------------------ | -------------- | --------------- |
> | Flux.1-dev  | 25.47             | 20.01              | 0.27           | 0.13            |
> | Kolors      | 24.71             | 18.90              | 0.20           | 0.15            |
> | **UNIMC**   | **93.55**         | **91.71**          | **30.01**      | **28.38**       |
>
> [Fig.4](https://ibb.co/Tq4WQWqv) presents a qualitative comparison of **UNIMC** with **Flux** and **Kolors**. As shown, the latter two models, relying solely on the prompt, fail to generate images that align with the provided keypoints.
>
> > Q4: Why our method outperforms prior work in keypoint-conditioned multi-class image generation?
>
> The core objective of our method is to enable keypoint-conditioned multi-class image generation.
> To this end, we introduce a **Unified Keypoint Encoder** that explicitly models the relationship between object categories and their corresponding keypoints, and a **Timestep-aware Keypoint Modulator** that enables fine-grained, keypoint-level feature modulation.
>
> In contrast to prior methods, our approach effectively addresses the challenges of **class binding fusion** and **keypoint binding fusion**, allowing the model to learn precise keypoint control across multiple classes simultaneously.
> As a result, **UNIMC achieves significant improvements over baseline methods** on this task.
>
> > Q5: Additional contributions beyond HAIG-2.9M and UNIMC.
>
> Our primary contribution is extending **keypoint-conditioned image generation** from a single class (human) to a **general multi-class setting**. To achieve this goal, we propose both the **HAIG-2.9M dataset** and the **UNIMC model**, which together enable **keypoint-conditioned multi-class image generation** across diverse object categories.

---

### Official Review · Reviewer_kiWJ · 2025-03-14

**Overall Recommendation:** 4

**Summary:**

The paper introduces UNIMC, a unified Diffusion Transformer framework for keypoint-guided multi-class image generation, and HAIG-2.9M, a large-scale dataset with 786K images and 2.9M instance annotations covering humans and 30 animal classes. UNIMC addresses limitations in existing approaches by using explicit class names, bounding boxes, and keypoint coordinates instead of skeleton images, which solves "class binding confusion" and "instance binding confusion" problems. The framework employs a unified keypoint encoder that maps different species' keypoints into a shared representation space and a timestep-aware keypoint modulator that injects keypoint tokens into the DiT backbone for fine control. Experiments demonstrate UNIMC outperforms previous methods like ControlNet and GLIGEN in image quality metrics, class accuracy, pose accuracy, and handling complex scenarios with multiple overlapping humans and animals.

**Claims And Evidence:**

The primary claims and their supporting evidence include:

1. **UNIMC's effectiveness for keypoint-guided generation**: The authors provide extensive quantitative metrics (FID, KID, CLIP scores, class accuracy, pose accuracy) in Tables 3 and 4 showing UNIMC outperforms baseline methods. This is reinforced by qualitative examples in Figures 6-8 that visually demonstrate the model's capabilities.

2. **Superiority of HAIG-2.9M dataset**: Table 1 provides a clear comparison with existing datasets, showing HAIG-2.9M's advantages in scale, diversity, and annotation quality. The benefits of training on this dataset are evidenced in Table 4, showing significant performance improvements compared to training on COCO, APT36K, or their combination.

3. **Effectiveness of the unified keypoint encoder and timestep-aware keypoint modulator**: The ablation studies in Table 7 methodically compare different configurations, showing Config a (the proposed approach) consistently outperforms alternatives across all metrics.

4. **Solution to class binding and instance binding confusion**: Qualitative examples in Figures 6-8 demonstrate the model's ability to correctly generate the appropriate class and manage multiple instances, especially in rows with overlapping subjects.

5. **Human preference**: Tables 8 and 9 provide human evaluation results that align with the quantitative metrics, strengthening the claim that UNIMC produces better results.

The only claim that could benefit from stronger evidence is the model's efficiency, which is mentioned but not extensively benchmarked in terms of computational requirements or training/inference times compared to alternatives.

**Essential References Not Discussed:**

N/A

**Experimental Designs Or Analyses:**

The experimental designs and analyses appear sound and appropriate. The paper uses relevant baselines (PIXART-α, ControlNet, GLIGEN), comprehensive metrics (image quality, alignment, class/pose accuracy), systematic ablation studies, cross-dataset comparisons, and human evaluations. The experiments effectively isolate component contributions and demonstrate UNIMC's advantages.

**Methods And Evaluation Criteria:**

The methods and evaluation criteria in this paper are well-aligned with the problem of keypoint-guided multi-class image generation.

**Methods:**
- Using explicit class names, bounding boxes, and keypoint coordinates instead of skeleton images directly addresses the identified limitations of previous approaches
- The DiT-based architecture is appropriate for high-quality image generation
- The unified keypoint encoder sensibly maps different anatomies into a shared representation space
- The HAIG-2.9M dataset is particularly well-suited as it directly addresses the identified gap in existing datasets by providing joint annotations for both humans and animals across diverse scenes.

**Evaluation Criteria:**
- Image quality metrics (FID, KID) are standard and appropriate
- Text-image alignment (CLIP) sensibly evaluates prompt faithfulness
- Class accuracy using YOLO-World effectively assesses whether correct subjects were generated
- Pose accuracy (AP) appropriately measures keypoint control fidelity
- Human preference studies validly capture subjective quality assessments
- Comparison against relevant baselines (ControlNet, GLIGEN) provides meaningful context
- Ablation studies effectively isolate component contributions

**Other Comments Or Suggestions:**

N/A

**Other Strengths And Weaknesses:**

## Other Weaknesses:

1. **Computational Requirements**: The paper lacks discussion of computational costs and efficiency. Information about training time, inference speed, and resource requirements would help assess practical applicability.
2. **Limited Diversity of Animal Species**: While 30 animal classes is a good start, it still represents a limited subset of the animal kingdom. Discussion of how the approach might generalize to unseen species would strengthen the paper.

**Questions For Authors:**

1. Could you provide details about the computational requirements of UNIMC compared to baseline methods? Specifically, what are the training time, memory requirements, and inference speed differences between UNIMC and alternatives like ControlNet and GLIGEN?

2. How well does the model generalize to unseen animal species not present in the HAIG-2.9M dataset? For instance, if provided with keypoints for a species like a kangaroo or platypus that wasn't in the training data, how would UNIMC perform?

3. Did you observe any systematic failure cases or limitations of UNIMC? Understanding typical failure modes would help assess the robustness of the approach.

4. The paper mentions using 8 A800 GPUs for training. What would be the minimum computational setup required to fine-tune UNIMC on a smaller dataset for a specific application?

5. How did you ensure the quality of keypoint annotations in HAIG-2.9M, particularly for animal species where keypoint definitions might be less standardized than for humans? What was the process for handling edge cases or ambiguous keypoint placements?

**Relation To Broader Scientific Literature:**

N/A

**Theoretical Claims:**

The paper doesn't contain any formal mathematical proofs or theoretical claims requiring verification.

---

> ### Author Rebuttal · Authors · 2025-03-30
>
> **Dear Reviewer kiWJ,**
>
> Thank you for your review and constructive comments. During the rebuttal period, we have made every effort to address your concerns. The detailed responses are provided below:
>
> > Q1: Details about the computational requirements of UNIMC compared to baseline methods.
> >
>
> Each method is trained using different datasets and GPU configurations, and their reported training costs are measured in different formats. For example, **ControlNet** reports GPU-hours, while **GLIGEN** uses steps * batch size. Therefore, directly comparing training costs across methods is not meaningful. In the table below, we report the **officially reported training costs of baseline methods**, along with our method’s training cost in both formats. Despite achieving significantly better performance than the baselines, our method maintains a relatively low training cost. For inference, all methods are evaluated on a **single RTX 3080 Ti** with **FP16 precision**, using a **50-step DDIM sampler** and **Flash-Attention-v2**. The generation resolution is **512×512**.
>
> Our method is more efficient than ControlNet, and although the inference time and memory usage are slightly higher than those of GLIGEN, the performance improvements over GLIGEN are substantial.
>
> | **Method** | **Training cost** | **Inference time (s/image)** | **Inference memory (GB)** |
> | --- | --- | --- | --- |
> | ControlNet | 300 GPU-hours with Nvidia A100 80G | 4.51 | 5.79 |
> | GLIGEN | Steps = 500K, Batch size = 32 | 4.01 | 4.90 |
> | UNIMC (Ours) | Steps = 8K, Batch size = 256 ≈ **576 GPU-hours** with Nvidia A800 80G | 4.23 | 5.08 |
>
> > Q2: How well does the model generalize to unseen animal species not present in the HAIG-2.9M dataset?
> >
>
> Although our model is trained on only **31 classes**, it generalizes well to unseen categories due to two key factors:
>
> 1. the **structural similarity across different species**, and
> 2. our use of a **text encoder** for class representation, which allows for extension to arbitrary categories.
>
> In addition, the underlying **pretrained T2I model** is inherently capable of class-to-image generation for a wide range of categories. Together, these components enable our method to theoretically scale to any class.
>
> Taking **kangaroo** as an example, we observe that its keypoint structure is similar to that of humans. Thus, we reused human keypoints to generate a kangaroo. As shown in [Fig.1](https://ibb.co/9mfwbtc8), the generated pose and spatial structure roughly align with the keypoints, suggesting that our model possesses a certain degree of generalization to **unseen species**.
>
> Therefore, we believe that as long as the training set covers a sufficient diversity of **structurally representative animal classes**, our model can generalize to a broad range of animal species.
>
> > Q3: Failure cases or limitations of UNIMC.
> >
>
> We observed that for some categories, a few keypoints are not perfectly controllable. For example, in the case of **cat** shown in [Fig.2](https://ibb.co/qLYjrjxn), the tail fails to be accurately generated. Beyond such rare cases, we did not observe significant failure cases in the test set.
>
> > Q4: What would be the minimum computational setup required to fine-tune UNIMC on a smaller dataset for a specific application?
> >
>
> Our use of 8 A800 GPUs was intended to accelerate training on a large-scale dataset. However, the UNIMC model itself is lightweight, with only **0.93B parameters** (and **0.35B trainable**). Under FP16 precision and batch size = 1, it requires only **12GB of GPU memory**.
>
> We fine-tuned UNIMC on a dataset of **1,000 cat images** using a **single RTX 3090**, with **batch size = 4** and **2K training steps**, taking about **4 hours**. The resulting **AP for the cat category reached 28.05**, and qualitative results are shown in [Fig.3](https://ibb.co/d4659mqZ).
>
> > Q5: How did you ensure the quality of keypoint annotations in HAIG-2.9M?
> >
>
> As described in Section 4.2, we first randomly sampled **5K images** and annotated them using several expert models. We then conducted a **user study** to select the best-performing model. As a result, our keypoint annotations reach the upper bound achievable by current pretrained models.
>
> Moreover, the animal species in our dataset are already included in the training data of the selected **pretrained keypoint estimator**. During annotation, we manually **sample and discard images with low-quality annotations**, and we also filter out images where keypoint predictions have **low confidence**.
>
> To further reduce **ambiguous keypoint placements**, we feed the  **bounding box** **detection results** into the keypoint estimator and **restrict keypoint detection to the corresponding bounding boxes.** This ensures that the estimator focuses only on the relevant regions, thereby improving annotation accuracy and reliability.

---

### Decision · Program_Chairs · 2025-05-01

**Decision:**

Accept (poster)

**Comment:**

Overall, the reviewers acknowledge the superiority of the proposed method and its effectiveness in addressing the target problem. While initial concerns were raised regarding the justification for the new dataset, the lack of model efficiency analysis, and limited comparisons with existing methods, these issues were largely addressed in the rebuttal. The reviewers were generally positive in their final recommendations. Although some concerns remain—particularly around comparison with related methods—they do not significantly impact the perceived contribution of this work.